# Numerical Simulation of the Effect of Different Footwear Midsole Structures on Plantar Pressure Distribution and Bone Stress in Obese and Healthy Children

**DOI:** 10.3390/bioengineering10111306

**Published:** 2023-11-10

**Authors:** Qixuan Zhou, Wenxin Niu, Kit-Lun Yick, Bingfei Gu, Yue Sun

**Affiliations:** 1School of Fashion Design & Engineering, Zhejiang Sci-Tech University, Hangzhou 310018, China; suyangsuhai@163.com (Q.Z.); gubf@zstu.edu.cn (B.G.); 2Shanghai Yang Zhi Rehabilitation Hospital, Tongji University School of Medicine, Shanghai 200125, China; niu@tongji.edu.cn; 3School of Fashion and Textiles, The Hong Kong Polytechnic University, Hong Kong; kit-lun.yick@polyu.edu.hk; 4Clothing Engineering Research Center of Zhejiang Province, Hangzhou 310018, China; 5Key Laboratory of Silk Culture Heritage and Products Design Digital Technology, Ministry of Culture and Tourism, Hangzhou 310018, China

**Keywords:** childhood obesity, finite element modeling, foot biomechanics, midsole structure

## Abstract

The foot, as the foundation of the human body, bears the vast majority of the body’s weight. Obese children bear more weight than healthy children in the process of walking and running. This study compared three footwear midsole structures (solid, lattice, and chiral) based on plantar pressure distribution and bone stress in obese and healthy children through numerical simulation. The preparation for the study included obtaining a thin-slice CT scan of a healthy 9-year-old boy’s right foot, and this study distinguished between a healthy and an obese child by applying external loadings of 25 kg and 50 kg in the finite element models. The simulation results showed that the plantar pressure was mainly concentrated in the forefoot and heel due to the distribution of gravity (first metatarsal, fourth metatarsal, and heel bone, corresponding to plantar regions M1, M4, and HM and HL) on the foot in normal standing. Compared with the lattice and solid EVA structures, in both healthy and obese children, the percentage reduction in plantar pressure due to the chiral structure in the areas M1, M4, HM, and HL was the largest with values of 38.69%, 34.25%, 64.24%, and 54.03% for an obese child and 33.99%, 28.25%, 56.08%, and 56.96% for a healthy child. On the other hand, higher pressures (15.19 kPa for an obese child and 5.42 kPa for a healthy child) were observed in the MF area when using the chiral structure than when using the other two structures, which means that this structure can transfer an amount of pressure from the heel to the arch, resulting in a release in the pressure at the heel region and providing support at the arch. In addition, the study found that the chiral structure was not highly sensitive to the external application of body weight. This indicates that the chiral structure is more stable than the other two structures and is minimally affected by changes in external conditions. The findings in this research lay the groundwork for clinical prevention and intervention in foot disorders in obese children and provide new research ideas for shoe midsole manufacturers.

## 1. Introduction

The phenomenon of obesity among children is serious in recent years; obesity is one of the main diseases threatening children’s health in today’s society and has become a global concern [1]. Obesity is detrimental to the development of children’s physical and mental health. It will not only cause a decline in athletic ability and cardiopulmonary dysfunction [2], affecting daily life, but also lead to abnormal skeletal muscle stress patterns [3], resulting in reduced stability in children’s posture, increasing the risk of injury and falls. The foot, as the foundation of the human body, bears the majority of the body’s weight [4]. Obese children bear more weight than healthy children in the process of walking and running. Moreover, due to the long-term overload in obese children, the physiological structure is prone to the collapse of the foot arch, leading to flat feet, foot inversion, knee valgus, etc. [5]. Increased mechanical loading can cause damage to the hip, knee ankle, and other joint parts, ultimately leading to issues such as injury, lesions, and lower-limb osteoarthritis. There is a significant difference in the distribution of plantar pressure and bone stress between obese children and healthy children during walking [6]. Obesity has a great influence on the arch structure, heel inversion, body stability, and impulse in children [7]. Footwear, as the intermediate medium between the foot and the ground, plays an important role in daily living and provides protection for the foot to avoid the impact of the ground, by reducing torque to achieve the effect of shock absorption [8]. At present, the common insole structure of children’s footwear is the solid structure of EVA elastic foam material [9,10]. This structure will not only aggravate the concentration of plantar stress in obese children but also lead to plantar fasciitis and foot ulcers caused by long-term uneven stress in obese children [11,12]. Compared with solid footwear, a hollow structure can evenly distribute the plantar pressure, also providing support for the foot arch and absorbing the ground reaction force. This structure could play the role of a shock buffer, reducing the internal and external rotation of the plantar fascia of in obese children [13,14].

The implications of obesity on the foot have been studied through gait analysis [13,15,16]. Previous studies of gait in obese subjects have shown that the greater the degree of obesity, the lower the postural stability and the more obvious the compensation by adjusting step width, temporal phase, and full-foot support time [17,18]. However, the results of all the previous studies on the soft tissues and structures of the foot in obese children (e.g., plantar pressure) were based on gait experiments that provide externally measurable parameters [13,15,16,17,18]. In contrast to these parameters, which can be measured by pressure transducers, internal stress cannot be obtained by experimental measurements since most foot pathologies caused by obesity (e.g., flat foot) are derived from within the foot bone to the soft tissue surface. They are difficult to identify and detect at an early stage, which can be addressed by finite element modeling (FEM). Few studies have been conducted to obtain information on internal stresses and strains in the foot skeleton through model simulations as a complement and validation of experimental results. FEM simulation techniques have been used as an effective analytical method to support biomechanical analysis of the human body and have great potential for application in the medical field over the years [19,20]. Furthermore, FEM allows biomechanical modeling and simulation of the soft tissue, bone, and midsole structures of the foot to analyze the deformation, force, displacement, and stress–strain of each tissue structure under different loading conditions, material properties, and boundary conditions [21,22,23,24].

Most of the previous studies on footwear midsole design via FE simulation were based on adult foot diseases, and few were designed for obese children to distribute the bone stress distribution and reduce plantar pressure. Ma et al. [25] used FE analysis to study the porous structure units of a diabetic insole designed with an adjustable gradient modulus, but they only focused on the porous structural units, and the foot model was not added. Xie et al. [26] used the FE method to design and optimize the shape of the sole for the elderly, which can realize the rapid customization of bionic sneakers for the elderly. However, their study did not consider the design of the shoe soles and only focused on the improvement of the shape of the insole. Jhou et al. [27] studied the deformation of the lattice structure when a 3D-printed polymer sandwich with a lattice core was placed in the midsole of the footwear but did not delve into the stress distribution in the foot bone and soft tissue. Li et al. [28] investigated the differences in peak plantar pressure during landing in the weight-bearing period between an FE model of a bare foot and a coupled FE model of the foot and barefoot running footwear, but only the condition of landing in the weight-bearing period was simulated using static analysis without considering the dynamic response while wearing the barefoot running footwear. Guo ying et al. [29] used FE static models to study the stress distribution in different shoe sole designs, such as Diamond, Grid, X shape, and Vintiles. However, they mainly focused on the design of the sole structure and lacked the analysis of the foot bones. Tang et al. [30] proposed a new design method that can generate a fully customized porous shoe sole and found that, compared with a flat shoe sole, the top surface of a customized sole could fully conform to the bottom surface of patient’s feet, which can significantly reduce peak plantar pressure. However, only the static loading condition was considered, and the functional volume was not manually divided into several sub-regions.

Therefore, this study aims to investigate the effect of different midsole structures on the foot biomechanics based on a static FE model. The effects of different footwear midsole constructions on plantar pressure and bone stress in healthy and obese children were also simulated and compared through FE analysis. The preliminary study on the analysis of plantar pressure and bone stress in obese children by FEM described in this paper can provide new ideas for manufacturers to research and develop footwear for obese children and give support to the clinical study of childhood flat foot caused by obesity with models.

## 2. Materials and Methods

### 2.1. Structural Design of Midsole

#### 2.1.1. Geometry Model of Midsole

The midsole of footwear is the structure between the insole and the outsole. Its functional characteristics are pressure reduction, cushioning, and rebound, reducing the damage caused by foot–ground impact, thus improving comfort and sports performance [31]. The design for midsole structures in this study included a solid midsole made of EVA material (solid EVA) and two hollow structures (chiral and lattice) with flexible resin material (E-shore A80) (Figure 1). The hollow structure is a cellular configuration made up of uniform and ordered unit cells [14]. The mechanical properties of hollow structures are determined by their architecture, such as cell type, cell size, and strut diameter. Based on the size of the subject’s foot, a midsole model of the form upper surface–structure–lower surface was constructed, in which the length, width, and height of the midsole structure were 240 mm × 180 mm × 34 mm, respectively, and the thickness of the upper and lower surface was 2 mm. The dimensions of the single-cell lattice in the midsole structure were 15 mm × 15 mm × 15 mm, and the wall thickness was 1.5 mm. The relative density (*ρ*) of the single-cell lattice and the number of cycles were kept consistent for the two hollow midsole structures so that the same amount of materials was used in 3D printing. The relative density *ρ* is defined as the ratio of the material volume (*V*_1_) to the total volume (*V*) (15 mm × 15 mm × 15 mm) of the single-cell lattice [32], which is an index used to represent the porosity of a structure. 

The chiral structure designed in this study was printed using a 3D printer (Envisiontec Perfactory^®^ P4 (EnvisionTEC, Gladbeck, Germany)), applying stereolithography (SLA) printing technology. The material chosen was flexible resin (E-shore A80) with density of 0.96 g/cm^3^ and a fracture tensile strength of 8.20 MPa.

#### 2.1.2. Compressive Behavior and Simulation

The compression test was carried out using an Instron 5566 universal mechanical test frame equipped with a 10 kN load, which was used to compress the surface of the structure (Figure 2). The compression test was conducted at 25 °C and a constant speed of 10 mm/min, which is in accordance with ASTM D575-91 Standard Test Methods [33].

The compressive behavior of the two hollow structures was simulated through FE modeling software (MSC Marc 2019) with quasi-static compression loading. The conditions imposed by FE were consistent with those of a compression experiment. The length, width, and height of the midsole structure used in the compression experiment were 75 mm × 75 mm × 34 mm, respectively. The mesh size in the FE simulations was 2 mm, and the material property of the midsole structure was assumed to be isotropic. The stress–strain curve of the chiral structure under a quasi-static loading condition in the test was compared with the results obtained using the FE model. Figure 3 shows the comparison between the stress–strain curves of the chiral structure obtained through numerical simulation and the experimental results. The root mean square error (RMSE) between them was 6.7%, which indicates that the simulation result is within an acceptable range. It can be seen that the magnitude and variation trend of the stress were in good agreement with the experimental results, which demonstrates the accuracy of the numerical method.

### 2.2. FE Model of Foot Structure

Static analysis of the FEM was used in this study to simulate the effects of different footwear midsole structures on plantar pressure and bone stress. The computational model incorporated geometrical models, material properties, and boundary conditions [34]. The flowchart of this study is shown in Figure 4.

#### 2.2.1. Geometric Modeling

The preparation for the study included obtaining a thin-slice CT scan of a healthy 9-year-old boy’s right foot (the subject has not had any previous foot surgery). The child was 137 cm tall; weighed 25 kg; and had normal, straight legs and feet. The data were obtained from Hangzhou children’s hospital, with an ethical approval document. The CT scan data were recorded into the 3D reconstruction software Mimics 21.0 to obtain the patient’s Dicom-format image, and the grayscale value was adjusted until the soft tissue and foot bone contours were displayed closest to the actual range. Then, the 3D model of the child’s foot bone was established through the image segmentation function. Geometrical processing was caried out on the 3D foot model obtained using Mimics software to wrap and smooth the surface of the geometry. The soft tissues around the foot bones were obtained through a Boolean operation. Then the 4-node tetrahedral element was meshed geometrically using the Hypermesh meshing module. The foot length and width of the 9-year-old child involved in this study were 22 cm and 16 cm, respectively. The constructed sub-model of the foot contained four parts: foot bones, soft tissues, plantar fascia, and Achilles tendons. Information on the element type, meshing size, and element numbers of each part is shown in Table 1. The anatomy of the foot bones is shown in Figure 5.

#### 2.2.2. Material Properties

The models used in this study were homogeneous, isotropic, and linear elastic except for the solid EVA. The Young’s modulus (E) and Poisson’s ratio (ν) for the materials included are shown in Table 2. A total of four contact FE models were constructed in this study, i.e., foot–chiral structure, foot–lattice structure, foot–solid EVA structure, and foot–ground.

#### 2.2.3. Loading and Boundary Conditions

For the boundary conditions, all the degrees of freedom of the distal tibia and fibula were fixed, and the distance between the foot and the structure was 1 cm. The body weight of a healthy child and an obese child were set to 25 kg and 50 kg in the FE simulation. Half of their body weights were transferred as the corresponding ground reaction forces applied to the bottom surface of the structures or ground in the upward direction to simulate a balanced standing position. Then, 75% of the body weight was loaded as the muscle force on the Achilles tendon. The friction between the surface of the midsole and the structure was set to 0.4 (Figure 6).

## 3. Results

### 3.1. Distribution of Plantar Pressure

The whole foot (HF) is divided into 10 regions according to the corresponding soft tissue of the foot bone [35,36], namely, the first toe (T1), the second to the fifth toe (T2~5), the first metatarsal bone (M1), the second metatarsal bone (M2), the third metatarsal bone (M3), the fourth metatarsal bone (M4), the fifth metatarsal bone (M5), the middle foot (MF), the medial heel (HM), and the lateral heel (HL) (Figure 7). The mean plantar pressure (MPP) and peak bone stress (PBS) are measured according to the 10 regions mentioned. In this study, the plantar pressure distribution and bone stress were compared in healthy and obese children among four different structures.

In order to compare the amount of MPP reduction before (barefoot standing on the ground can be regarded as the comparison group) and after wearing the shoe midsole (standing on various midsole structures), the percentage reduction (*Pr*) was defined as below:$$P_{ri}=\frac{p_{0}-p_{i}}{p_{0}}$$
*i* = 1, 2, 3, *P*_1_ = *P_lattice_*, *P*_2_ = *P_Chiral_*, *P*_3_ = *P_solid EVA_*, *P*_0_ = *P_ground_*

A positive value of this item means the MPP of the region is reduced, whereas a negative value indicates that the MPP increased.

As is shown in Figure 8 and Table 3, the *Pr* of the MPP in the M4 region brought about by the lattice midsole structure (*Pr_1-M4_*) was −7.16% for obese children and −15.97% for healthy children. A reduction of −109.58% for an obese child and −99.18% for a healthy child was observed in the MPP in the MF region. For other foot regions, in both obese and healthy children, the values of *Pr* for the lattice structure were all positive, which indicates plantar-pressure-releasing effects in these areas. For the chiral structure, the MPP increased in the M5 region and the MF region with a *Pr* of −196.41% and −172.98% for obese children and −171.03% and −369.73% for healthy children, respectively. Other than these two regions, the remaining parts were noted to have a decrease in MMP in both obese and healthy children on wearing the chiral midsole structure. Moreover, in the MF region, the increase in the MPP for the chiral structure was greater than that for the lattice structure in both obese and healthy children. For the solid EVA structure, the MPP increased slightly in the M1 (−2.73%) and HL (−4.17%) regions for obese children, while it showed a mild rise in the M1 (−7.61%) and HL (−2.63%) regions for healthy children. The M4 region had the highest MPP with a *Pr* of −37.41% (obese) and −27.88% (healthy). The MPP of other foot regions decreased after wearing the solid EVA midsole structure. 

Figure 9 and Table 4 show the mean plantar pressure distribution in the 10 regions (T1, T2~5, M1, M2, M3, M4, M5, MF, HM, and HL) in obese and healthy children when the bare foot presses on the ground and the different midsole structures. The uniform contour bars in Figure 9a,c represent the MPP distribution through the whole foot area. As the range of plantar pressure was different along different foot areas, separated contour bars for the different regions (Figure 9b,d) were adopted to compare the MPP of various foot areas among different midsole structures. In terms of contact pressure across the plantar region, the MPP in HL was greater than that in HM in both obese and healthy children. The regions with greater differences in plantar pressure distribution were all concentrated in M1, M3, M4, HM and HL, while those with smaller differences were in T1, T2~5, M2, M5 and MF. Lower contact pressure occurred in M2 and M5 than in M1, M3, and M4 in both obese and healthy children, with M1 having the greatest plantar pressure [25] (42.73 kPa and 34.44 kPa for lattice, 35.64 kPa and 32.21 kPa for chiral, and 59.71 kPa and 39.06 kPa for solid EVA). The MPP of T2~5 (0.51 kPa and 0.29 kPa for lattice, 0.73 kPa and 0.34 kPa for chiral, and 0.94 kPa and 0.46 kPa for solid EVA) was greater than that of T1 (0.42 kPa and 0.26 kPa for lattice, 0.36 kPa and 0.17 kPa for chiral, and 0.49 kPa and 0.20 kPa for solid EVA) for all three structures in obese and healthy children.

### 3.2. Stress Distribution of Foot Bone

The results for the maximum von Mises stress (MVSS) of the foot bone and the *Pr* of the MVSS relative to the comparison group (ground) in obese and healthy children with different midsole structures are shown in Table 5, Table 6, and Figure 10, respectively. As can be seen in Figure 10, in both obese and healthy children, the MVSS for the chiral structure was reduced at the lateral cuneiform with a *Pr* of 26.88% for an obese child and 19.16% for a healthy child; at the medial cuneiform with a *Pr* of 5.60% for an obese child and 12.50% for an healthy child; at the first metatarsal with a *Pr* of 13.02% for an obese children and 5.62% for a healthy children; and at the third to the fifth metatarsal with *Pr* values of 26.85%, 34.12%, 39.04% (obese children) and 3.96%, 30.67%, 49.58% (healthy children), respectively. The MVSS for the lattice structure was reduced at the fourth and fifth metatarsal with a *Pr* of 23.65% and 42.12% for obese children and 29.07% and 28.57% for healthy children. The MVSS for the solid EVA was reduced at the navicular with a *Pr* of 2.04% for an obese child and 3.72% for a healthy child; at the lateral cuneiform with a *Pr* of 1.05% for an obese child and 2.09% for a healthy child; at the medial cuneiform with a *Pr* of 0.51% for an obese child and 0.36% for a healthy child; at the first to the fifth metatarsal with a *Pr* of 0.55%, 2.07%, 0.64%, 1.81%, and 5.48% for an obese child and 0.80%, 2.20%, 1.62%, 2.40%, and 3.78% for a healthy child; and at the first to the third phalange with *Pr* values of 0.80%, 0.77%, 0.20% (obese child) and 1.59%, 1.67%, 1.26% (healthy child), respectively. Among them, the MVSS of the above three structures in both healthy and obese children was reduced in the fourth metatarsal and fifth metatarsal, while the chiral structure had the greatest *Pr* at the fourth metatarsal compared with the lattice and the solid EVA structures.

## 4. Discussion

### 4.1. Comparison of Distribution of Plantar Pressure

Figure 11 presents the distribution of plantar pressure in 50 kg obese children and 25 kg healthy children with different midsole structures. It can be seen that the plantar pressure was mainly concentrated in the metatarsal region and the heel region, with almost no pressure on the arch part. In contrast, the hollow structures (chiral and lattice) transferred some of the pressure from the heel to the arch when compared with the solid EVA and the comparison group (ground), resulting in a release in the pressure at heel region and providing support at the arch. A larger area of contact surface was observed for the chiral structure in the arch and heel than for the lattice, resulting in an even distribution of body weight and providing more support to the arch. 

In simulations of the three structures with the comparison group, the pressure in the lateral heel region (HL) was significantly greater than that in the medial heel region (HM), and the pressure in the lateral phalange region (T2~5) was significantly greater than that in the medial phalange region (T1) in both obese and healthy children.

There are three important pressure points in the plantar fascia that distribute the pressure on the foot [37], i.e., first metatarsal, fourth metatarsal, and heel bone, corresponding to the plantar regions M1, M4, and HM and HL. The heel bone bears the majority of the gravitational force and the impulse due to the change in the COG (center of gravity), which acts on the heel bone and soft tissue surrounding the heel bone [38]. In a 25 kg healthy child, the *Pr* in the M1, M4, HM, and HL regions was 33.99%, 28.25%, 56.08%, and 56.96% for the chiral structure; 29.42%, −15.97%, 37.12%, and 35.14% for the lattice structure; and 19.97%, −27.88%, 21.66%, and −2.63% for solid EVA, respectively. In a 50 kg obese child, the *Pr* in the M1, M4, HM, and HL regions was 38.69%, 34.25%, 64.24% and 54.03% for the chiral structure; 26.50%, −7.16%, 24.04%, and 36.88% for the lattice structure; and −2.73%, −37.41%, 4.07%, and −4.17% for solid EVA, respectively. It was found that, in both healthy and obese children, the *Pr* of plantar pressure in the M1, M4, HM and HL regions for the chiral structure was greater than that for the lattice and solid EVA structures, which could largely relieve the pain in the heel and forefoot parts caused by gravity concentrated at three pressure points during walking. Moreover, the *Pr* of plantar pressure in HM and HL was greater than that in M1 and M4 for the three midsole structures.

### 4.2. Comparison of Stress Distribution of Foot Bone

Of the 17 foot bones in a healthy child, solid EVA had more bones with a higher *Pr* of MVSS (15 bones in total, all bones except the talus and intermediate cuneiform) than the lattice (2 bones in total, fourth and fifth metatarsal) and chiral (6 bones in total, lateral cuneiform, medial cuneiform, first metatarsal, and third to fifth metatarsal) structures, but the value of *Pr* for the chiral structure was greater than that for the lattice and solid EVA structures. Of the 17 foot bones in an obese child, the lattice structure had more bones with a higher *Pr* of MVSS (12 bones in total, all bones except the calcaneus, talus, intermediate cuneiform, cuboid, and fifth phalange) than the chiral (9 bones in total, medial cuneiform, lateral cuneiform, intermediate cuneiform, first metatarsal, third to fifth metatarsal, and first and second phalange) and solid EVA (11 bones in total, all bones except the calcaneus, talus, intermediate cuneiform, cuboid, and fourth and fifth phalange) structures, but the value of *Pr* for the chiral structure was greater than that for the lattice and solid EVA structures.

In both healthy and obese children, the MVSS of the calcaneus was the greatest, followed by that of the talus, for all three structures. The reason for this may be due to the way the boundary conditions are imposed, resulting in increased bone stress around the tibia and fibula due to fixation of the distal tibia and fibula.

### 4.3. Comparison of Obese and Healthy Child

It could be seen that a greater *Pr* occurred for the chiral structure in the area in which the three pressure points are located than for the other two structures in both obese and healthy children. Moreover, the greater the weight, the more pronounced the effect of the reduction and the more even the distribution of the plantar pressure, which could reduce the impact force from the ground.

For the lattice structure, the *Pr* of MPP increased in T1, T2~5, M2, M5, and HL in an obese child compared with a healthy child, with a *Pr* value of 59.10%, 55.55%, 12.10%, 62.27%, and 36.88% in obese children and 56.89%, 50.06%, 7.10%, 57.06%, and 35.14% in healthy children. For the chiral structure, the *Pr* of MPP increased in M1, M4, and HM in an obese child compared with a healthy child, with a *Pr* value of 38.69%, 34.25%, and 64.24% in obese children and 33.99%, 28.25%, and 56.08% in healthy children. For the solid EVA structure, the *Pr* of MPP increased in M2, M5, and MF in an obese child compared with a healthy child, with a *Pr* value of 29.14%, 42.62%, and 46.81% in obese children and 25.15%, −7.61%, and 8.04% in healthy children. It was found that the chiral structure was not highly sensitive to the external application of body weight, which means that the region and value of the *Pr* will not change much with an increase in body weight. This indicates that the chiral structure was more stable than the other two structures and was minimally affected by changes in external conditions. It was found that the support effect and stress dispersion effect of the chiral structure in obese children were more obvious than those in healthy children. In addition, the results commonly show that, among the three structures, the chiral structure had the best performance in plantar pressure distribution and bone stress distribution.

However, this study has some limitations. The geometric model of the foot was overly simplified, neglecting ligaments, muscles, and nerves. In addition, this study also lacked model validation corresponding to the FE model. Future work should, therefore, consider more-precise model construction and validation. Furthermore, it was found that the chiral structure was minimally affected by changes in external conditions, but the changes in external conditions in this paper were limited to two different values of body weight (50 kg and 25 kg), and future research can introduce more values of body weight.

## 5. Conclusions

This study compared three numerical models of different footwear midsole structures with a static analysis on plantar pressure distribution and bone stress in obese and healthy children. The simulation results of the three different footwear midsole structures were also compared in terms of the plantar pressure distribution and bone stress distribution.

It was found that the plantar pressure was mainly concentrated in the forefoot and heel due to the distribution of gravity (three pressure points) on the foot in normal standing. For the chiral structure, the plantar pressure was transferred from the heel to the arch when compared with the solid EVA and the comparison group (ground), resulting in a release in the pressure at the heel region and providing support at the arch. A larger area of contact surface was observed for the chiral structure in the arch and heel than for the other three midsole structures, resulting in an even distribution of body weight and providing more support to the arch. 

In addition, the chiral structure was not highly sensitive to the external application of body weight, that is, the region and value of *Pr* did not change much with an increase in body weight. This indicates that the chiral structure was more stable than the other two structures and was minimally affected by changes in external conditions.

The validated FEM model discussed in this study could be used to predict foot deformation and contact pressure in obese children and provide new research ideas for shoe midsole manufacturers. Moreover, it could also provide FE support for clinical prevention and intervention in foot disorders in obese children, such as flat foot, horseshoe foot, and other pathologic foot conditions.

## Figures and Tables

**Figure 1 bioengineering-10-01306-f001:**
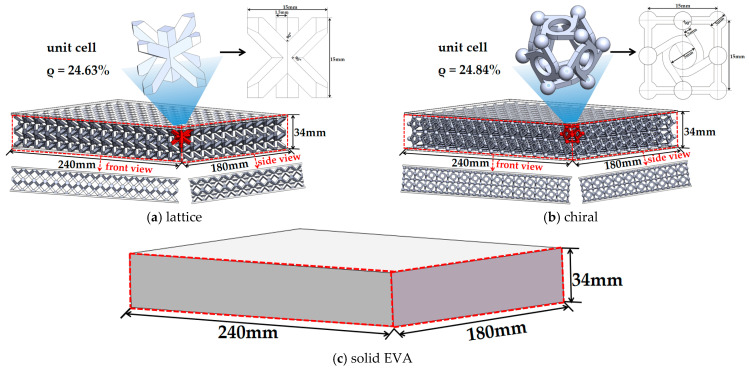
Design for midsole structures: (**a**) lattice, (**b**) chiral, and (**c**) solid EVA.

**Figure 2 bioengineering-10-01306-f002:**
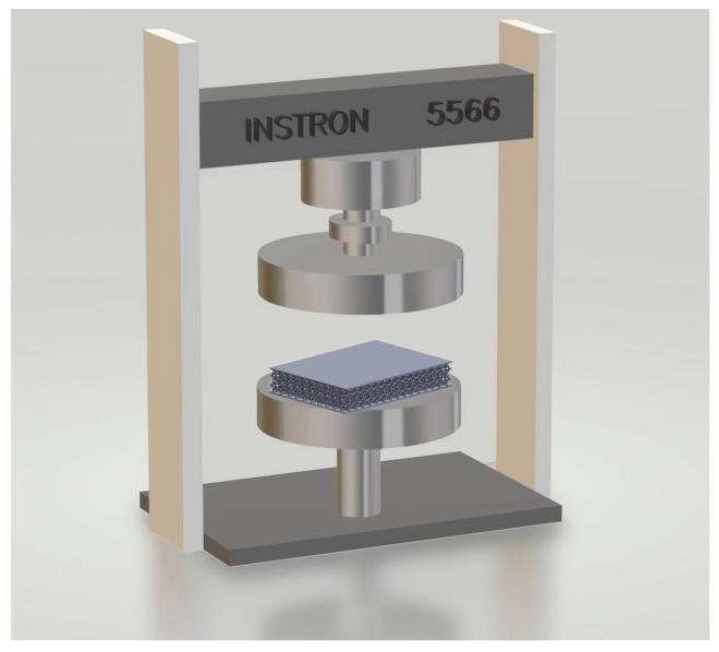
Instron 5566 universal mechanical test frame.

**Figure 3 bioengineering-10-01306-f003:**
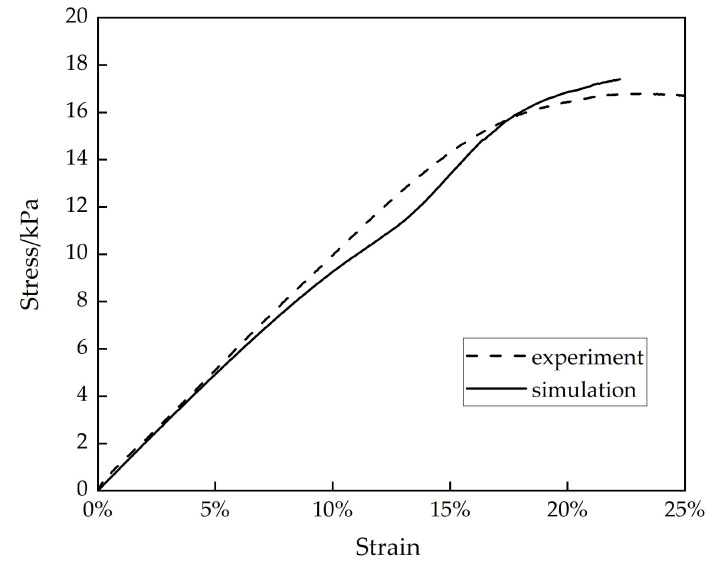
Stress–strain curves.

**Figure 4 bioengineering-10-01306-f004:**
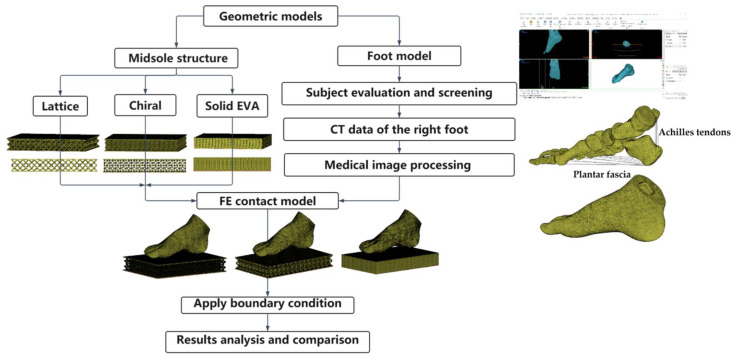
Flowchart of entire study.

**Figure 5 bioengineering-10-01306-f005:**
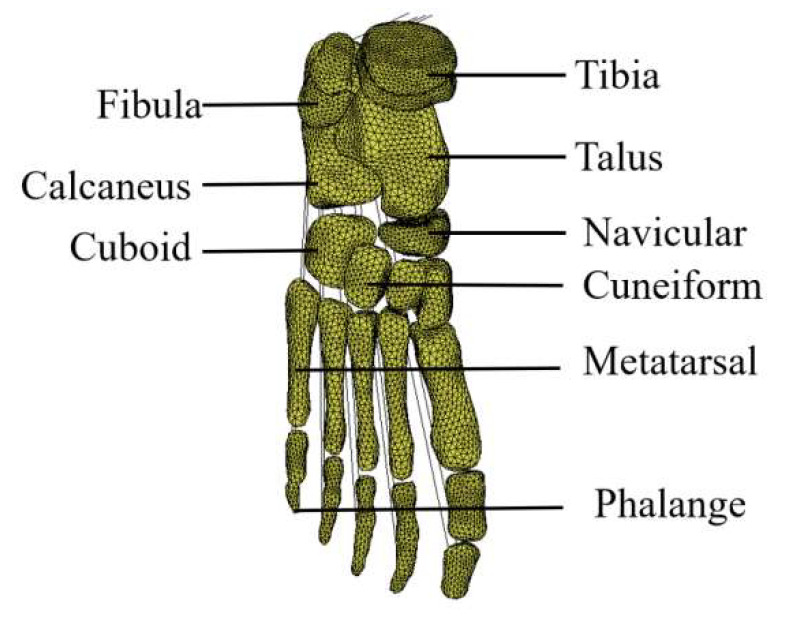
Foot bone anatomy.

**Figure 6 bioengineering-10-01306-f006:**
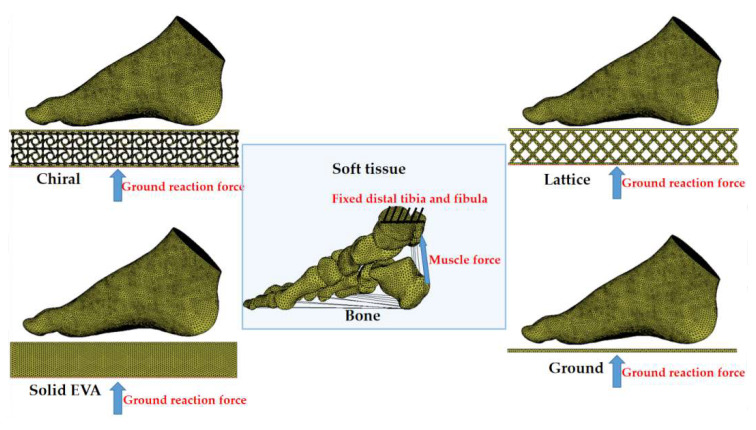
Loading and boundary conditions in FE models.

**Figure 7 bioengineering-10-01306-f007:**
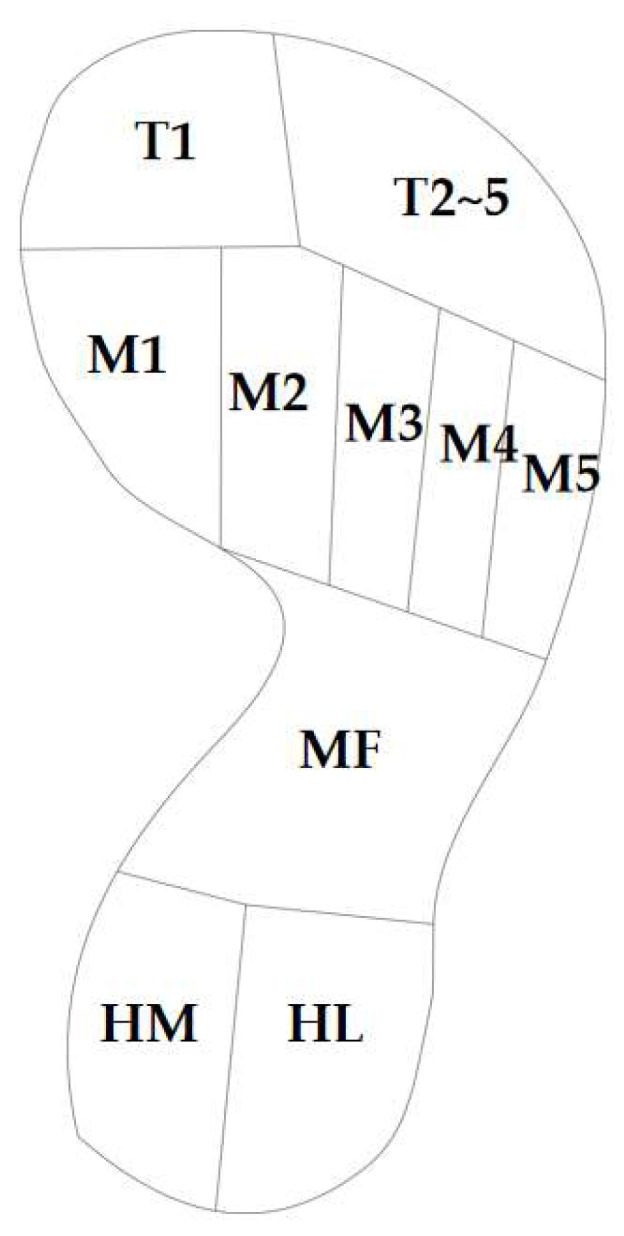
Ten regions of the whole foot (HF).

**Figure 8 bioengineering-10-01306-f008:**
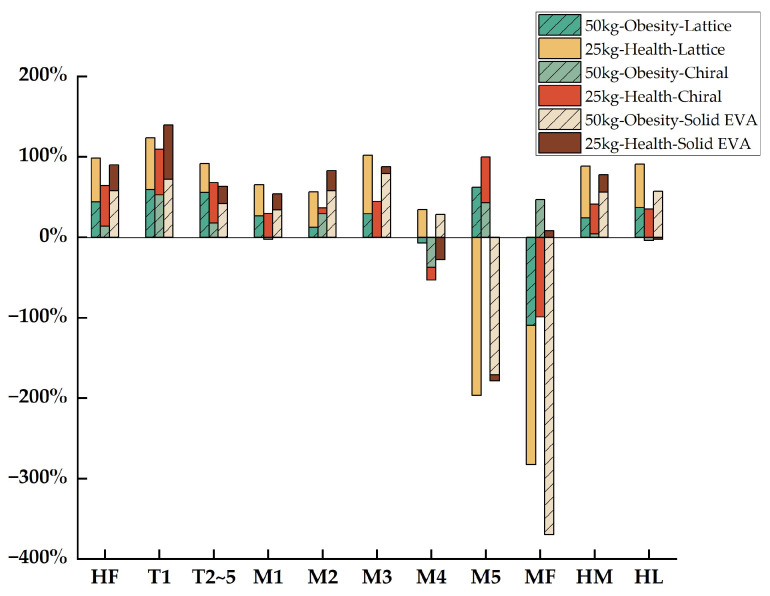
*Pr* of the MPP of three structures (lattice, chiral, and solid EVA) relative to the comparison group (ground).

**Figure 9 bioengineering-10-01306-f009:**
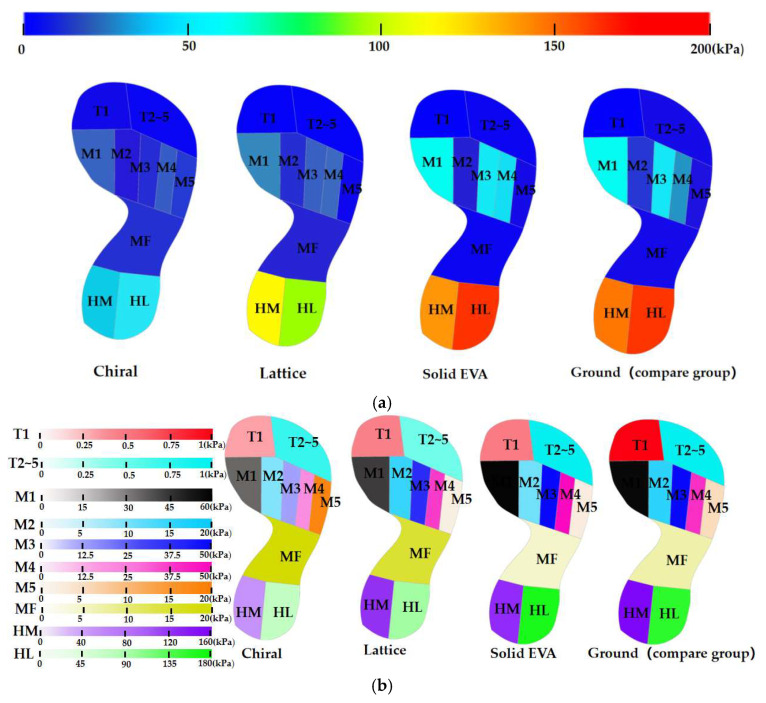
Mean plantar pressure distribution in 10 regions (T1, T2~5, M1, M2, M3, M4, M5, MF, HM, HL) in obese and healthy children on different structures: (**a**) Distribution of MPP for 50 kg children with uniform contour bars. (**b**) Distribution of MPP for 50 kg children with different contour bars. (**c**) Distribution of MPP for 25 kg children with uniform contour bars. (**d**) Distribution of MPP for 25 kg children with different contour bars.

**Figure 10 bioengineering-10-01306-f010:**
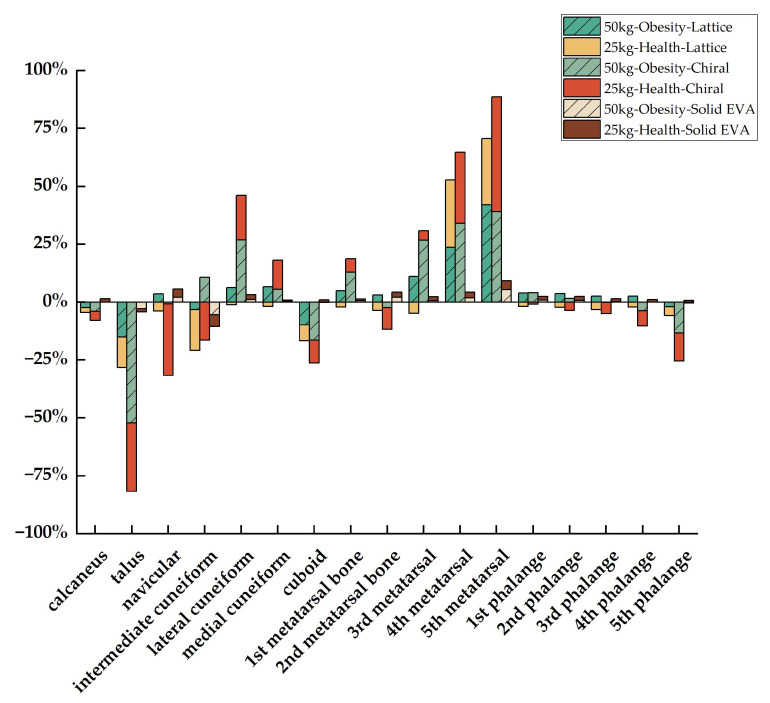
*Pr* of the MVSS of three structures (lattice, chiral, and solid EVA) relative to the comparison group (ground).

**Figure 11 bioengineering-10-01306-f011:**
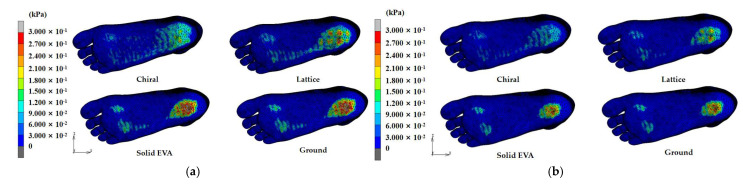
Distribution of plantar pressure in obese and healthy children with different structures: (**a**) 50 kg obese children and (**b**) 25 kg healthy children.

**Table 1 bioengineering-10-01306-t001:** Information on element type, meshing size, and element numbers of foot structure.

Part	Element Type	Meshing Size (mm)	Element Number
Bone	3D Tetrahedral	1.5	107,545
Soft tissue	3D Tetrahedral	1.5	478,783
Plantar fascia	Line element	2	10
Achilles tendons	Line element	2	4

**Table 2 bioengineering-10-01306-t002:** Material properties of FE model.

Component	Material Type	*E* (MPa)	*ν*
Soft tissue	Linear elasticity	0.45	0.49
Bone	Linear elasticity	7300	0.3
Lattice/Chiral	Linear elasticity	8.9	0.33
Achilles tendon	Linear elasticity	816	0.3
Plantar fascia	Linear elasticity	3.5	0.4
Ground	Linear elasticity	210,000	0.3
Solid EVA	Foam	μ_1_ = 2.13, μ_2_ = −1.012, α_1_ = 11.95, α_2_ = −6.06, β_1_ = 0, β_2_ = 0	

**Table 3 bioengineering-10-01306-t003:** *Pr* of MPP in obese and healthy children with different midsole structures.

	*Pr* of MPP/%					
	50 kg Obese Child			25 kg Healthy Child		
	Lattice	Chiral	Solid EVA	Lattice	Chiral	Solid EVA
HF	43.73%	54.55%	13.49%	50.53%	57.87%	32.03%
T1	59.10%	64.54%	52.48%	56.89%	72.13%	67.21%
T2~5	55.55%	36.25%	17.83%	50.06%	41.72%	21.28%
M1	26.50%	38.69%	−2.73%	29.42%	33.99%	19.97%
M2	12.10%	44.24%	29.14%	7.10%	57.62%	25.15%
M3	29.27%	72.70%	0.14%	44.44%	78.94%	8.68%
M4	−7.16%	34.25%	−37.41%	−15.97%	28.25%	−27.88%
M5	62.27%	−196.41%	42.62%	57.06%	−171.03%	−7.61%
MF	−109.58%	−172.98%	46.81%	−99.18%	−369.73%	8.04%
HM	24.04%	64.24%	4.07%	37.12%	56.08%	21.66%
HL	36.88%	54.03%	−4.17%	35.14%	56.96%	−2.63%

**Table 4 bioengineering-10-01306-t004:** MPP in obese and healthy children with different midsole structures.

	Mean Plantar Pressure/kPa							
	50 kg Obese Child				25 kg Healthy Child			
	Lattice	Chiral	Solid EVA	Ground	Lattice	Chiral	Solid EVA	Ground
HF	50.07	40.44	76.97	88.97	22.58	19.23	31.02	45.64
T1	0.42	0.36	0.49	1.02	0.26	0.17	0.20	0.61
T2~5	0.51	0.73	0.94	1.14	0.29	0.34	0.46	0.59
M1	42.73	35.64	59.71	58.13	34.44	32.21	39.06	48.80
M2	14.86	9.42	11.98	16.90	12.24	5.59	9.87	13.18
M3	36.48	14.08	51.50	51.58	22.12	8.38	36.35	39.81
M4	38.78	23.80	49.73	36.19	20.97	12.98	23.13	18.09
M5	2.29	17.99	3.48	6.07	1.11	6.99	2.78	2.58
MF	11.66	15.19	2.96	5.56	2.30	5.42	1.06	1.15
HM	116.34	54.77	146.91	153.15	50.10	35.00	62.42	79.68
HL	108.74	79.19	179.47	172.28	61.29	40.67	96.98	94.49

**Table 5 bioengineering-10-01306-t005:** MVSS of foot bone in obese and healthy children with different midsole structures.

	Maximum Von Mises Stress/kPa							
	50 kg Obese Child				25 kg Healthy Child			
	Lattice	Chiral	Solid EVA	Ground	Lattice	Chiral	Solid EVA	Ground
Calcaneus	5788	5885	5656	5656	3018	3068	2912	2956
Talus	2512	3322	2246	2183	1259	1441	1127	1112
Navicular	1043	1089	1059	1081	559	704	518	538
Intermediate cuneiform	867	750	886	840	532	526	475	452
Lateral cuneiform	537	419	567	573	290	232	281	287
Medial cuneiform	367	371	391	393	285	245	279	280
Cuboid	879	931	801	800	430	443	399	403
1st metatarsal bone	343	314	359	361	254	235	247	249
2nd metatarsal bone	1030	1087	1040	1062	518	547	489	500
3rd metatarsal	974	801	1088	1095	582	533	546	555
4th metatarsal	423	365	544	554	266	260	366	375
5th metatarsal	169	178	276	292	170	120	229	238
1st phalange	604	603	623	628	385	381	372	378
2nd phalange	1369	1399	1410	1421	856	868	824	838
3rd phalange	1454	1493	1489	1492	898	913	859	870
4th phalange	1633	1738	1676	1676	991	1035	960	971
5th phalange	894	994	880	877	533	575	509	513

**Table 6 bioengineering-10-01306-t006:** *Pr* of MVSS in obese and healthy children with different midsole structures.

	*Pr* of MVSS/%					
	50 kg Obese Child			25 kg Healthy Child		
	Lattice	Chiral	Solid EVA	Lattice	Chiral	Solid EVA
Calcaneus	−2.33%	−4.05%	0.00%	−2.10%	−3.79%	1.49%
Talus	−15.07%	−52.18%	−2.89%	−13.22%	−29.59%	−1.35%
Navicular	3.52%	−0.74%	2.04%	−3.90%	−30.86%	3.72%
Intermediate cuneiform	−3.21%	10.71%	−5.48%	−17.70%	−16.37%	−5.09%
Lateral cuneiform	6.28%	26.88%	1.05%	−1.05%	19.16%	2.09%
Medial cuneiform	6.62%	5.60%	0.51%	−1.79%	12.50%	0.36%
Cuboid	−9.88%	−16.38%	−0.13%	−6.70%	−9.93%	0.99%
1st metatarsal bone	4.99%	13.02%	0.55%	−2.01%	5.62%	0.80%
2nd metatarsal bone	3.01%	−2.35%	2.07%	−3.60%	−9.40%	2.20%
3rd metatarsal	11.05%	26.85%	0.64%	−4.86%	3.96%	1.62%
4th metatarsal	23.65%	34.12%	1.81%	29.07%	30.67%	2.40%
5th metatarsal	42.12%	39.04%	5.48%	28.57%	49.58%	3.78%
1st phalange	3.82%	3.98%	0.80%	−1.85%	−0.79%	1.59%
2nd phalange	3.66%	1.55%	0.77%	−2.15%	−3.58%	1.67%
3rd phalange	2.55%	−0.07%	0.20%	−3.22%	−4.94%	1.26%
4th phalange	2.57%	−3.70%	0.00%	−2.06%	−6.59%	1.13%
5th phalange	−1.94%	−13.34%	−0.34%	−3.90%	−12.09%	0.78%

## Data Availability

Data are contained within the article.

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
