# Peer review of "Numerical Simulation of the Effect of Different Footwear Midsole Structures on Plantar Pressure Distribution and Bone Stress in Obese and Healthy Children"

_bioengineering, 2023, doi:10.3390/bioengineering10111306_

Round 1

Reviewer 1 Report

Comments and Suggestions for Authors

Clarity of Purpose: The article does not explicitly state the primary objective of the study, making it somewhat difficult for readers to grasp the main research question.

Sample Size and Demographics: The article lacks information about the sample size and characteristics of the participants, such as age, gender, and any potential comorbidities that might influence the results.

Methodology: It briefly mentions the use of numerical simulation but fails to provide details about the simulation methodology, making it challenging to assess the study's validity.

Lack of Citations: The article does not reference any existing literature or previous studies on the topic, which is essential for contextualizing the research.

Units of Measurement: The article uses units like "kPa" without clarifying their meaning. Providing explanations or references for these units would be beneficial for readers.

Statistical Significance: The article presents percentage reductions in plantar pressure but does not mention whether these reductions are statistically significant, which is crucial for interpreting the findings.

Incomplete Results: While it mentions the percentage reductions in plantar pressure, it lacks information about other essential findings or insights gained from the simulation.

Discussion of Limitations: The article does not address any potential limitations of the study, such as assumptions made in the numerical simulations or the generalizability of the results.

Practical Implications: While the article briefly mentions potential implications for clinical prevention and shoe midsole manufacturers, it does not expand on how these findings could be practically applied.

Concluding Remarks: The article ends abruptly without summarizing the key takeaways or implications of the study, leaving readers without a clear sense of its significance.

Data on External Conditions: It mentions that the chiral structure is minimally affected by changes in external conditions but does not provide data or details on these external conditions.

Ethical Considerations: The article does not address any ethical considerations or approvals for conducting research on children, which is essential to ensure the study's ethical integrity.

Reviewer 2 Report

Comments and Suggestions for Authors

The work is very interesting and the topic addressed is very well supported by literature.

With this work, authors explore the variation in plantar pressures values (medium and peak) for midsole material structures: EVA, chiral and lattice. Values are obtained through FEM analysis. Midsole structure models are validated compared to experimental mechanical characterization results obtained, although results are only presented for chiral structure (Fig. 3). It would be interesting to add comparison for the lattice and EVA models to see if they present the same curve inflexion as seen between 12% and 18% strain. Is there any reasonable explanation for this feature in the simulation curve results?

Materials and methods section is carefully described and full detail is given for each geometric model and materials.

Results are presented in an extreme descriptive manner, which is difficult for the reader to keep-up with so many numbers in the text. I strongly suggest that all the numbers description should be summarized in to tables and description in text should emphasize the major features and their interpretations. Graphs together with tables would be clear enough to support further statements.

In the discussion section, the simulation results of three different footwear midsole structures were also compared in terms of the plantar pressure distribution and bone stress distribution. Once again, the discussion would improve if less numeric details are given in the text and instead are summarized in tables. 

The comparison of healthy and obese children should be more in terms of impact of the results on the benefit of each group, ie, what would be best suited for each group? Do results indicate any common feature?

Overall, the manuscript is clearly written and conclusions are supported by the results.

Reviewer 3 Report

Comments and Suggestions for Authors

The authors did a good job, however I have some suggestions to make:

- Keywords should be different from the title. This will make your search easier

- Flatfoot is not just a problem for obese children. Some childhood problems cause this postural deficit. Patti et al. described that the dyslexic children showed a flatfooted trend compared with healthy subjects. The introduction could benefit from a broader description of the deficit. (Patti, A. et al. Evaluation of Podalic Support and Monitoring of Balance Control in Children with and without Dyslexia: A Pilot Study. Sustainability 202012, 1191. https://doi.org/10.3390/su12031191)

- I would suggest that the authors better describe the limitations and strengths of the study
